# Prevalence and Antibiotic Resistance Pattern of *Streptococcus*, *Staphylococcus*, *Neisseria meningitidis* and *Enterobacteriaceae* in Two Reference Hospitals of Yaoundé: An Overview before and during COVID-19 Pandemic Era

**DOI:** 10.3390/antibiotics12050929

**Published:** 2023-05-18

**Authors:** Cecile Ingrid Djuikoue, Willy Yamdeu Djonkouh, Cavin Epie Bekolo, Rodrigue Kamga Wouambo, Raspail Carrel Founou, Paule Dana Djouela Djoulako, Gilder Tonfak Temgoua, Benjamin D. Thumamo Pokam, Nicolas Antoine-Moussiaux, Teke R. Apalata

**Affiliations:** 1Department of Public Health, Faculty of Health Sciences, University of Montagnes, Bangangte 00237, Cameroon; yamdeuwilly@gmail.com; 2American Association of Microbiology (ASM), ASM Cameroon, Bangangte 00237, Cameroon; 3Department of Epidemiology and Public Health, Faculty of Medicine and Pharmaceutical Sciences, University of Dschang, Dschang 00237, Cameroon; 4Foundation of Epidemiological Surveillance of Biological Germs, Douala 00237, Cameroon; 5Division of Hepatology, Department of Medicine II, Leipzig University Medical Center, University of Leipzig, 04103 Leipzig, Germany; 6Department of Microbiology-Hematology and Immunology, Faculty of Medicine and Pharmaceutical Sciences, University of Dschang, Dschang 00237, Cameroon; 7Antimicrobial Research Unit, School of Health Sciences, College of Health Sciences, University of KwaZulu-Natal, Durba 4001, South Africa; 8Antimicrobial Resistance and Infectious Diseases (ARID), Research Institute of Centre of Expertise and Biological Diagnostic of Cameroon (CEDBCAM-RI), Yaoundé 00237, Cameroon; 9Faculty of Medicine, Sorbonne University, 75013 Paris, France; 10Deido District Hospital, Douala 00237, Cameroon; 11Faculty of Health Sciences, University of Buea, Buea 00237, Cameroon; 12Fundamental and Applied Research for Animals and Health, Faculty of Veterinary Medicine, University of Liege, 4000 Liège, Belgium; 13Faculty of Health Sciences, Walter Sisulu University, Mthatha 5099, South Africa

**Keywords:** resistant bacteria, COVID-19 pandemic era, antibiotics

## Abstract

The COVID-19 pandemic led to tremendously use of antimicrobial due to the lack of proper treatment strategies, raising concerns about emergence of antimicrobial resistance (AMR). This study aimed at determining the prevalence and antibiotic resistance pattern of selected bacteria isolates in 02 referral health facilities in Yaoundé before and during the COVID-19 pandemic era. We conducted a retrospective study over a period of 03 years (from 1 January 2019 to 31 December 2021) in the bacteriology units of the Central and General Hospitals of Yaoundé, Cameroon. Data on bacteria genera (*Streptococcus*, *Staphylococcus*, *Neisseria meningitidis* and *Enterobacteriaceae*) as well as their corresponding specifics antibiotics: Cefixime, azythromycin and erythromycin were obtained from laboratory records. The global resistance rate of bacteria as well as their correlation with antibiotics according to COVID-19 pandemic era was determined and compared. For *p* < 0.05, the difference was statistically significant. In all, 426 bacterial strains were included. It appeared that the highest number of bacteria isolates and lowest rate of bacterial resistance were recorded during the pre-COVID-19 period in 2019 (160 isolates vs. 58.8% resistance rate). Conversely, lower bacteria strains but greater resistance burden were recorded during the pandemic era (2020 and 2021) with the lowest bacteria amount and peak of bacteria resistance registered in 2020, the year of COVID-19 onset (120 isolates vs. 70% resistance in 2020 and 146 isolates vs. 58.9% resistance in 2021). In contrast to almost all others groups of bacteria where the resistance burden was quite constant or decreasing over years, the *Enterobacteriaceae* exhibited greater resistance rate during the pandemic period [60% (48/80) in 2019 to 86.9% (60/69) in 2020 and 64.5% (61/95) in 2021)]. Concerning antibiotics, unlike erythromycin, azythromycin related resitance increased during the pandemic period and the resistance to Cefixim tends to decrease the year of the pandemic onset (2020) and re-increase one year therafter. A significant association was found between resistant *Enterobacteriaceae* strains and cefixime (R = 0.7; *p* = 0.0001) and also, between resistant *Staphylococcus* strains and erythromycin (R = 0.8; *p* = 0.0001). These retrospective data showed a herogeneous MDR bacteria rate and antibiotic resistance pattern over time before and during the COVID-19 pandemic era suggesting that antimicrobial resistance needs to be more closely monitored.

## 1. Introduction

Antibiotic resistance is rising to dangerously high levels in all parts of the world. New resistance mechanisms are emerging and spreading globally, threatening our ability to treat common infectious diseases [1]. In February 2017, the World Health Organization (WHO) published its first list of priority pathogens resistant to antibiotics, thus requiring the research and development of new molecules [2]. The list in question enumerates the 12 families of bacteria most threatening to human health [2]. This work by WHO is part of its efforts to combat growing antimicrobial resistance (AMR) around the world [2]. It is a new tool to ensure that research and development meets urgent public health needs. The most critical group includes multidrug resistant (MDR) bacteria that pose a particular threat in hospitals, or for patients whose care requires the use of devices such as respirators or blood catheters [2]. It includes Acinetobacter producing a carbapenemase, *Pseudomonas* producing a carbapenemase and various *Enterobacteriaceae* resistant to carbapenems and beta-lactams [2]. They are often involved in fatal infections, such as blood infections and pneumonia. The second and third groups (the high and medium priority categories) include other increasingly resistant bacteria causing more common illnesses such as gonorrhea or *salmonella* food poisoning. Whereas priority 2 (high) consists of germs such as in *Staphylococcus aureus* (Methicillin or vancomycin resistant); *Neisseria gonorrhoeae*, (cephalosporin or fluoroquinolone resistant) [2], priority 3 (medium) is made up of *Streptococcus pneumoniae* (penicillin resistant); *Haemophilus influenzae* (ampicillin resistant); *Shigella* spp. (resistant to fluoroquinolones) [2].

Bacteria are talked to be multi-resistant (MDR) to antibiotics when, due to the accumulation of acquired resistance to more than 03 families of antibiotics, they are no longer sensitive to more than a small number of antibiotics that can be used therapeutically [3]. The high frequency, the pathogenic potential and the commensal nature expose MDR bacteria to the risk of their dissemination outside the hospital [4]. One hand, Antimicrobial resistance (AMR) can be ancient and it is considered in this case as the expected result of the interaction of many organisms with their environment [5]. Indeed, most antimicrobial compounds are naturally-produced molecules, and, as such, co-resident bacteria have evolved mechanisms to overcome their action in order to survive [5]. Other hand, in clinical settings, we are typically referring to the expression of “acquired resistance” in a bacterial population that was originally susceptible to the antimicrobial compound [6]. This can be the result of mutations in chromosomal genes or the acquisition of external genetic determinants of resistance, likely obtained from intrinsically resistant organisms present in the environment [6].

The consumption of antibiotics during COVID-19 increased tremendously [7,8] especially during the onset of the pandemic and the most prescribed antibiotics were: Amoxicillin, cefixime, imipenem, azythromycin and erythromycin [9]. In low and middle income countries (LMICs), the lack of awareness about disease outbreaks, misconceptions, misinformation, cultural stigma surrounding the virus and fear to get infected encourage individuals to self-prescribe potent antimicrobials such as azithromycin, doxycycline and even local traditional medicines without fully comprehending the dangers [10]. Taken together, these factors raised concern about emergence of antimicrobial resistance (AMR). Indeed, access to effective antimicrobials is largely unavailable in LMICs, while rates of AMR are expected to expand 4–7 times faster [11]. Additionally, because of high rates of improper antibiotic prescribing for COVID-19 patients, treatment interruptions for persons with chronic illness, and broad use of antimicrobial drugs by local populations, COVID-19 has most possibly increased the rate of AMR-related consequences [12]. It therefore sounds really interesting as from now on to investigate the resistance of bacteria to antibiotics. The aim of this study was to determine the prevalence of *Streptococcus*, *Staphylococcus*, *Neisseria meningitidis* and *Enterobacteriaceae* and their resistance pattern to either Cefixime, azythromycin or erythromycin before and during the COVID-19 pandemic era.

## 2. Results

### 2.1. Age and Sex of the Study Population

From 2019 to 2021, 426 strains were isolated from patients attending the bacteriology unit (a single bacterial strain per patient) of 02 referral hospitals for COVID-19 including 160 in 2019, 120 in 2020 and 146 in 2021.

Among those 426 bacterial strains, we recorded 67 females vs. 93 males in 2019; 51 females vs. 69 males in 2020 and finally 66 females vs. 80 males in 2021. The age range (25–50) years was predominant during the 3 years (53.1% in 2019, 43.3% in 2020 and 49.3% in 2021) and the mean age of the study population was 54.8 ± 18.9 years in 2019, 40.3 ± 15.5 in 2020 and 38 ± 14.6 in 2021. The most common sample was the pus, with 61.9% (see Table 1).

### 2.2. Bacteria Profile of the Study Population

From the Figure 1, we noticed that 51.4% (219/426) of bacteria isolates were *Enterobacteriaceae*; 45% (192/426) of *Staphylococcus* strains; 3% (13/426) of *Streptococcus* strains and 0.3% (2/426) of *Neisseria meningitidis*.

### 2.3. Number of Bacteria Isolate and Their Corresponding Resistance Rate over Years

The horizontal bar chart and curves below (Figure 2) show the numbers of bacterial strains and their resistance rate over years.

From these graphs (Figure 2), it tends to appear that the highest number of bacteria isolates and lowest rate of bacterial resitance were recorded (160 isolates vs. 58.8% resistance rate) during pre-COVID-19 period (in 2019). However, we noticed in contrast lower bacteria isolates and greater resistance burden during COVID-19 pandemic era (year 2020 and 2021). The lowest bacteria amount and peak of bacteria resistance were registered in 2020, the year of COVID-19 onset (120 isolates vs. 70% resistance in 2020 and 146 isolates vs. 58.9% resistance in 2021)

### 2.4. Resistance Pattern of Bacterial Strains Isolated in 2019, 2020 and 2021

#### 2.4.1. Resistance Rate by Group of Bacteria

Figure 3 showed that unlike almost all others groups of bacteria where the resistance burden was quite constant or slightly decreasing over years, the *Enterobacteriaceae* exhibited greater resistance rate during the pandemic period [60% (48/80) in 2019 vs. 86.9% (60/69) in 2020 and 64.2% (61/95) in 2021]. Furthermore, no resitance to antibiotics has been noted concerning *Neisseria meningetidis* in 2020 and 2021 [100% (1/1) in 2019 vs. 0% of strain in 2020 and 2021].

The graph below represents the resistance rate by group of bacteria over time.

#### 2.4.2. Global Resistance Rate of Bacteria according to the Antibiotics of Interest

Cefixime, azythromycin, erythromycin were antibiotics of interest in this study. The Figure 4 shows the distribution of bacterial resistance to these antibiotics.

This graph highlights that overall bacterial resistance rate to azythromycin increased during the pandemic period with the highest rate of resistance recorded in 2020 (4.6% resistance rate in 2019, 15.5% in 2020 and 6.9% in 2021) over time. The resistance to Cefixim tends to decrease the year of the pandemic onset (2020) and re-increase one year therafter (60.6% resistance rate in 2019, 36.9% in 2020 and 70.9% in 2021) and concerning erythromycin, its resistance rate was slightly decreasing over time (19.1% resistance rate in 2019, 14.3% in 2020 and 10.5% in 2021).

#### 2.4.3. Resistant Rate according to the COVID-19 Era and Year of Isolation

The table below reflects the rate of bacterial resistance with regard to year of isolation (Table 2).

From that table, the bacterial resistance rate globally increased in 2020 and 2021 with the highest rate of resistance more likely recorded in 2020, the year of COVID onset (*p* = 0.05).

#### 2.4.4. Bacteria Resistant Rate before and during COVID-19 Pandemic Era according to Gender

The table below reflects the rate of bacterial resistance according to gender before and during COVID-19 pandemic period (Table 3).

The table reveals an increase bacterial resistance rate in female subjects over years [56.7 in 2019, 74.5% in 2020 and 78.9% in 2021] with a significantly higher bacteria resistance rate in female than males in 2021 [(78.9% vs. 42.5%, *p* = 0.0000, OR [95%CI]: 5.03 [2.40–10.51]].

In this study, the data basically showed a significant association between gender and bacterial resistance rate [69.6% (128/184) females vs. 56.2% (136/242) males, *p* = 0.05, OR [95%CI]: 1.78 (1.19–2.67)]. However, the chi-square of interaction or confounding did not show any statistically significant difference (*p*_(int)_ = 0.2 and p_(MH)_ = 0.4 respectively).

#### 2.4.5. Age and Global Resistance Rate according to COVID-19 Pandemic Era

The table below shows the rate of bacterial resistance according to age before and during COVID-19 pandemic period (Table 4).

In comparison to subjects aged (0–25) with decreasing resistance rate over time, the subjects aged (25–50) and >50 more likely exhibited relatively high resistant bacteria with the peak of resistance recorded in 2020, the year of COVID-19 onset. However, the observed differences were not statistically significant.

### 2.5. Repartition of Bacterial Resistant Rate to Selected Antibiotics before (2019) and during COVID-19 Pandemic (2020 and 2021)

The Table 5 below highlights relationships between each group of resistant bacteria and the selected antibiotic before and during the COVID-19 pandemic period.

This table reveals that taken individually, resistance of each bacteria isolate to its selected antibiotic tends to slightly decrease the year of the COVI-19 pandemic onset (2020) and then, either re-increase or stay constant one year thereafter (2021). We noted significantly relationships between resistant bacterial strains and some specific antibiotics over years, such as between cefixime and resistant *Enterobacteriaceae* strain [(R = 0.5; *p*= 0.0001 in 2019); (R = 0.7; *p* = 0.0001 in 2020); (R = 0.5; *p* = 0.0001 in 2021)]; then between erythromycin and resistant *Staphylococcus* [(R = 0.4; *p* = 0.0001 in 2019); (R = 0.8; *p* = 0.0001 in 2020); (R = 0.7; *p* = 0.0001 in 2021)] and *Streptococcus* strains [(R = 0.0.2; *p* = 0.01 in 2019); (R = 0.4; *p* = 0.0005 in 2021)]; Similarly, *Streptococcus* resistance were linked to the use of azythromycin just in 2019 (R = 0.2; *p* = 0.04).

## 3. Discussion

In order to determine the prevalence of *Streptococcus*, *Staphylococcus*, *Neisseria meningitidis* and *Enterobacteriaceae* and their resistance pattern to either Cefixime, azythromycin or erythromycin before and during the COVID-19 pandemic era, a retrospective study was conducted in 02 referral health facilities in Yaounde, Cameroon. For that purpose, data were collected from each hospital databases (laboratory records) before and during the pandemic (in 2019, 2020 and 2021 respectively), then, analyzed thereafter.

In all, 426 strains were isolated from patients attending the bacteriology unit of the 02 referral hospitals from 2019 to 2021 as follow: 37.5% (160/426) in 2019, 28.2% (120/426) in 2020 and 37.27% (146/426) in 2021. These results coincide to those of Chih-Cheng and al in patients of Veterans General Hospital of Taiwan who also found a decreasing number of several bacteria species (*Streptococcus pneumonia*, *Staphylococcus aureus*, *Enterococcus* spp. and *Klebsiella pneumonia*) from 2019 to 2020 [9]. The high number of isolated strains in the pre-CoVID19 period (2019) contrasting with its lowest amount in the year of COVID onset (in 2020) is logical and could rely on the wide and systematic implementation of COVID-19 barriers measures including hygienic measures, wearing of face mask, social distancing that lead to rapid and overall decline of respiratory tract infections as well as feco-oral transmitted diseases [13]. In fact, giving similar routes of transmission, all the barriers measures applicable for COVID-19 have the advantage to also block others infections spread [14]. Furthermore, the observed decline in bacteria isolates in hospital settings during COVID-19 period could also be due to the significant drop in attendances for non-COVID-19-related conditions [15,16,17].

Concerning the bacterial profile, 51.4% (219/426) were *Enterobacteriaceae*, 45% (192/426) *Staphylococcus* strains, 3% (13/426) *Streptococcus* strains and 0.3% (2/426) of *Neisseria meningitides*. Chih-Cheng also found *Enterobacteriaceae* as the most isolated bacteria 30.8% (49/159) in patients at Veterans General Hospital of Taiwan between 2019 and 2020 [9]. Previous studies worldwide have also reported the predominance of *Enterobacteriaceae* in clinical samples [8,9,10,11,12,13,14,15,16,17,18,19,20,21,22,23]. In fact, *Enterobacteriaceae* are the most incriminated commensal germs in opportunistic infections in general [9,21]. In this study, the main isolated species among *Enterobacteriaceae* were *Klebsiella pneumoniae* 41.1% (90/219), *E. coli* 35.6% (78/219), *Enterobacter* 11.4% (25/219) and *Proteus mirabilis* 9.13% (20/219), similar trends has been already observed in previous studies [9,22]. In fact, the rapid emergence of bacteria especially resistant bacteria is occurring worldwide, endangering the efficacy of antibiotics, which have transformed medicine and saved millions of lives [23,24,25,26].

In this study, the bacterial resistance tends to increase in female’s subjects over years and subjects aged (25–50) and >50 presented a relative high rate of resistance with likely a peak of resistance in 2020, the year of COVID-19 onset (*p* > 0.005). Previous studies already reported a high prevalence of bacterial resistant strain either with ageing [21] and or in female’s subjects [4]. This disproportionate burden of antimicrobial resistance (AMR) on women is due to both demand and supply-side factors. Some demand-side factors which increase women’s vulnerability to AMR are biological factors, women’s nature and type of employment, excessive home-based care work, and limited access to healthcare [27]. On the supply-side, gender differences in antibiotic prescription by doctors due to lack of training and gender-bias increase women’s antibiotic usage (AMU) [28,29]. In fact, women are 27% more likely to receive an antibiotic prescription in their lifetime compared to men [30]. Furthermore, the high rate of AMR in subject aged >25 years during the COVID-19 pandemic in this study is not surprising. This aged range corresponds to the most active part of the Cameroonian population who is believed to have taken the highest amount of medications including antibiotics to protect against COVID-19 at work as the pandemic evolved in 2020.

This study also showed in the pre-COVID-19 period (2019) that the highest number of bacteria isolates and lowest rate of bacterial resistance were more likely recorded (160 isolates vs. 58.8% resistance rate). However, we noticed in contrast lower bacteria isolated during COVID-19 pandemic era but greater resistance burden with the lowest bacteria amount and peak of bacteria resistance registered in 2020, the year of COVID-19 onset (120 isolates vs. 70% resistance in 2020 and 146 isolates vs. 58.9% resistance in 2021). A similar study has already reported unexpected high incidence of infections due to MDR bacteria among COVID-19 patients admitted to the intensive care unit [14]. Moreover, a metanalysis of 1331 articles of Ruwandi shows that during the first 18 months of the pandemic, AMR prevalence was high in COVID-19 patients and varied by hospital and geography although there was substantial heterogeneity [8]. According to Ruwandi et al., the increase in the prevalence of resistant strains in 2020 could be straightly related to the peak of antibiotic consumption during the COVID-19 pandemic onset, ranging up to 74.7% from 2019 to 2020 for antibiotics such as fosfomycin [8]; that generally increased the selection pressure and evolvement of resistance mechanism [8].

In constrast to almost all others groups of bacteria where the resistance burden was quite constant or decreasing over years, *Enterobacteriaceae* exhibited greater resistance rate during the pandemic period [60% (48/80) in 2019 vs. 86.9% (60/69) in 2020 and 64.2% (61/95) in 2021)]. Furthermore, no resitance to antibiotics has been noted concerning *Neisseria meningetidis* in 2020 and 2021 [100% (1/1) in 2019 vs. 0% of strain in 2020 and 2021)]. A similar study on the impact of SARS-CoV-2 epidemic on AMR shows that, the Gram-negative bacteria were isolated from 78% of patients with predominant *Enterobactericaea* resistant strains including carbapenemases producers such as *K. pneumoniae* (92.6%) and A. baumannii (72.8%) [9]. This result is not different from that of Sanofi, which estimates that 30% to 51% of *Klebsiella pneumoniae* strains show resistance to 3rd generation cephalosporins in Poland, from 2017 to 2021 [18]. Unlike *Neisseria meningetidis*, *Enterobacteriaceae* are widely distributed and have a large host range [31], they can cross-infect and spread between medical staff and patients, and also their genetic materials (such as plasmids or transposons) can be obtained from the outside world, leading to horizontal transmission of drug-resistant genes, which further leads to the wide spread of drug-resistant bacteria [32,33].

The resistance of each bacterium isolate to its selected antibiotic tends to slightly decrease the year of the COVI-19 pandemic onset (2020) before re-increasing or staying constant one year thereafter (2021). A significant association was found between resistant *Enterobacteriaceae* strains and cefixime (R = 0.7; *p* = 0.0001) and also, between resistant *Staphylococcus* strains and erythromycin (R = 0.8; *p* = 0.0001). Chih-Cheng et al. reported in 2020 Taiwan a rapid increase in multidrug-resistant organisms (MDROs), including extended-spectrum β-lactamase (ESBL)-producing Klebsiella pneumoniae, carbapenem-resistant New Delhi metallo-β-lactamase (NDM)-producing *Enterobacterales*, *Acinetobacter baumannii*, methicillin-resistant *Staphylococcus aureus* (MRSA) [9]. Besides, several recent reports have described an increase in multidrug-resistant organisms (MDROs) during the COVID-19 pandemic [33,34,35,36,37]. The cause of this high resistance according to Rawson et al. is multifactorial and could be particularly related to high rates of utilization of antimicrobial agents in COVID-19 patients with a relatively low rate of co- or secondary infections [38]. Indeed, AMR should be continuously closely monitored during and after COVID-19 pandemic era.

## 4. Materials and Methods

### 4.1. Study Duration and Location

A retrospective cross-sectional and analytical study was conducted at the Central Hospital and the General Hospital of Yaounde from September 2021 to July 2022. Data from laboratory records were collected over a period of 3 years from 1 January 2019 to 31 December 2021.

### 4.2. Sampling Method and Sampling Population

In this study, the retrospective data (from 2019 to 2021) of patients suffering from a bacterial infections caused by one of the 4 bacteria genera (*Streptococcus*, *Staphylococcus*, *Neisseria meningitidis* and *Enterobacteriaceae)* and their susceptibility test results to the 3 following antibiotics (Cefixime, azythromycin and erythromycin) were collected from laboratory records at the Central and General hospitals of Yaoundé. We included only patients’data with complete information about the patient name and address, age, gender, the type of sample, the dates of sampling and analysis, the techniques used for analysis, the bacteria isolated (only *Streptococcus*, *Staphylococcus*, *Neisseria meningitidis* and/or *Enterobacteriaceae)*, the susceptibility testing to either cefixime, azythromycin or erythromycin of each selected bacteria isolates according to EUCAST recommendations. This procedure was validated by the regional ethics committee for Center region Cameroon (CE N°02229/CRERSHC/2022), the Institutional ethic Committee for Research on Human Health of the Yaounde General Hospital (Ref N°22322/HGY/DG/DPM/APM-TR) and Yaounde Central Hospital (N°2022/121/AR/MINSANTE/SG/DHCY/UAF).

### 4.3. Microbiological Methods

All participating reference hospitals strictly complied with the standard operating procedures for intended samples collection and culture. According to the guidelines of the Antibiogram Committee of the French Society of Microbiology (CA-SFM/EUCAST), local experienced laboratory members of each hospital independently completed the isolation, identification of isolates, and antibiotic susceptibility testing. In brief, the colonies obtained were firstly subjected to macroscopic examination (description of the size, color, and appearance of the colonies), followed by a further identification and antibiotic susceptibility testing of a bacterial suspension (0.5 McFarland diluted to 1.5 × 10^7^ CFU/mL in 0.45% saline) using the VITEK 2 System (VITEK^®^ 2, BioMerieux, France). The minimal inhibitory concentrations (MIC) were determined for all tested antibiotics and results were interpreted according to the guidelines of Antibiogram Committee of the French Society of Microbiology (CA-SFM/EUCAST) [39]. In this study, only the antibiogram results of cefixime, azythromycin or erythromycin of the 4 bacteria genera of concern were considered.

### 4.4. Statistical Analysis

Data were collected using Kobotolbox software, entered into a Microsoft Excel 2016 database (version 15.13.3) and analyzed using StatView software (version 5.0.0.0). A simple linear regression was used to show the relationships between the antibiotics of interest and the resistant bacterial strains. Moreover, the Chi-Square of independence to compare bacterial resistance rate in 2019 (pre-COVID-19 period) and 2020/2021 (COVID-19 period).

Data such as age, sex, location of patients were presented as a percentage whereas frequency of isolated bacteria, their resistance profile to selected antibiotics were presented in diagram and tables. Furthermore, a chi-square of interaction was calculated and an assessment of confounding and interaction using stratified analysis was performed by the Mantel-Haenszel risk estimation method.

## 5. Conclusions

As compared to the pre-COVID-19 era, these retrospective data showed an increase of MDR bacteria rate during the COVID-19 onset (2020) follow by a decrease one year thereafter (2021). Unlike age of participant, gender was associated to MDR bacteria rate in 2021. Concerning bacteria genera, *Enterobacteriaceae* exhibited greater resistance rate during the pandemic period suggesting that antimicrobial resistance still needs to be more closely monitored.

## Figures and Tables

**Figure 1 antibiotics-12-00929-f001:**
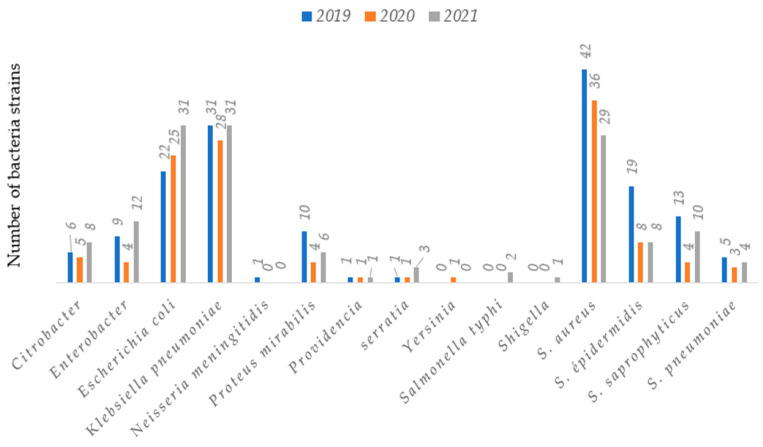
Breakdown of bacteria species over years.

**Figure 2 antibiotics-12-00929-f002:**
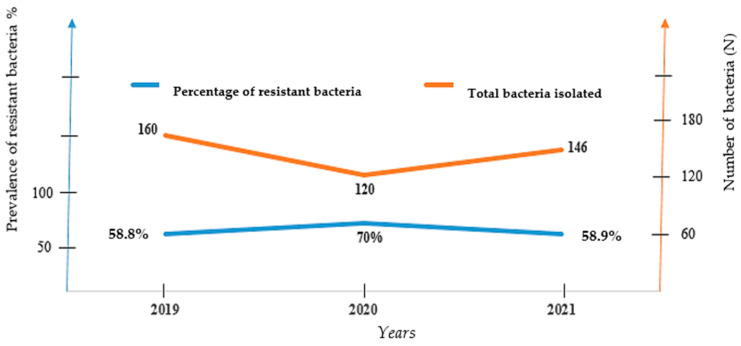
Bacteria isolates and resistance rate over time.

**Figure 3 antibiotics-12-00929-f003:**
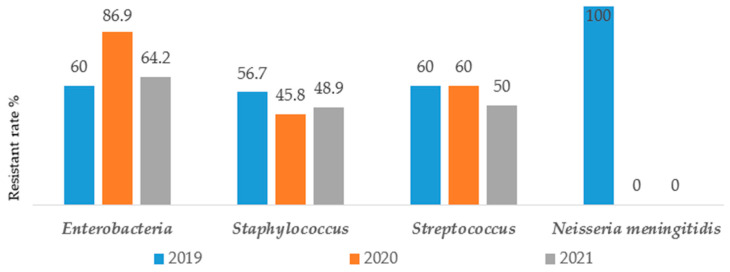
Resistance rate by group of bacteria in 2019, 2020 and 2021.

**Figure 4 antibiotics-12-00929-f004:**
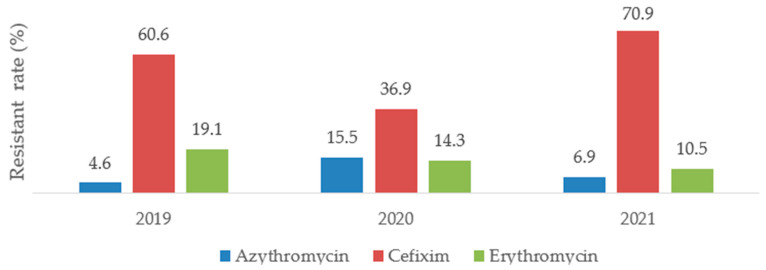
Resistance rate to antibiotics according to COVID-19 pandemic period.

**Table 1 antibiotics-12-00929-t001:** Age and sex of the study population.

			*n*	%
Pre-COVID19 (2019) *n* = 160	Sex	Male	93	38.4
		Female	67	36.4
	Age	(0–25)	30	18.7
		(25–50)	85	53.1
		>50	45	28.1
During COVID-19 (2020) *n* = 120	Sex	Male	69	28.5
		Female	51	27.7
	Age	(0–25)	31	25.8
		(25–50)	52	43.3
		>50	37	30.8
During COVID-19 (2021) *n* = 146	Sex	Male	80	33
		Female	66	35.9
	Age	(0–25)	37	25.3
		(25–50)	72	49.3
		>50	37	25.3

**Table 2 antibiotics-12-00929-t002:** Resistant rate according to year of isolation.

		Bacterial Resistance Rate			
	Year	Yes *n* (%)	No *n* (%)	χ^2^	*p*-Value
Pre-COVID-19	2019	94 (58.7)	66 (41.2)	Ref	Ref
During COVID-19	2020	84 (70)	36 (30)	3.75	0.05
	2021	86 (58.9)	60 (41.1)	0.001	0.97

Ref: reference. Interaction chi-square χ^2^_int_ (1ddl) = 1.29, *p*_(int)_ = 0.2; Mantel Haenzel Chi-square χ^2^_MH_ = 0.42, *p*_(MH)_ = 0.4.

**Table 3 antibiotics-12-00929-t003:** Bacteria resistant rate according to gender.

		Bacterial Resistance Rate			
		Yes *n* (%)	No *n* (%)	χ^2^	*p*-Value
Pre-COVID19 (2019) (*n* = 160)	Male	56 (60.2)	37 (39.8)	Ref	Ref
	Female	38 (56.7)	29 (43.3)	0.2	0.66
During COVID-19(2020) *n* = 120	Male	46 (66.7)	23 (33.3)	Ref	Ref
	Female	38 (74.5)	13 (25.5)	0.86	0.35
During COVID-19(2021) *n* = 146	Male	34 (42.5)	46 (57.5)	Ref	Ref
	Female	52 (78.9)	14 (21.2)	19.67	**0.0000**

Ref: reference. Interaction chi-square χ^2^_int_ (1ddl) = 1.29, *p*_(int)_ = 0.2; Mantel Haenzel Chi-square χ^2^_MH_ = 0.42, *p*_(MH)_ = 0.4.

**Table 4 antibiotics-12-00929-t004:** Bacterial Resistance rate according to age and the pandemic era.

		Bacterial Resistance Rate			
		Yes *n* (%)	No *n* (%)	χ^2^	*p*-Value
Pre-COVID-19 (2019) *n* = 160	(0–25)	22 (73.3)	8 (26.7)	Ref	Ref
	(25–50)	44 (51.8)	41 (48.2)	4.21	0.04
	>50	28 (62.2)	17 (37.8)	1	0.3
During COVID-19 (2020) *n* = 120	(0–25)	19 (61.3)	12 (38.7)	Ref	Ref
	(25–50)	38 (73)	14 (27)	1.25	0.26
	>50	27 (73)	10 (27)	1.05	0.30
During COVID-19 (2021) *n* = 146	(0–25)	18 (48.6)	19 (51.4)	Ref	Ref
	(25–50)	45 (62.5)	27 (37.5)	1.92	0.16
	>50	23 (62.4)	14 (38)	1.34	0.24

Ref: reference. Interaction chi-square χ^2^_int_ (1ddl) = 1.29, *p*_(int)_ = 0.2; Mantel Haenzel Chi-square χ^2^_MH_ = 0.42, *p*_(MH)_ = 0.4.

**Table 5 antibiotics-12-00929-t005:** Correlation between each group of resistant bacteria and its specific antibiotic over time.

Bacterial Strains	Overall Resistance	Cefixime Resistance	Azythromycin Resistance	Arythromycin Resistance	R	*p*-Value
**2019**						
*Enterobacteriaceae*	48 (60.00%)	34 (70.83%)	-	-	0.5	0.0001
*Staphylococcus*	42 (56.75%)	-	7 (16.66%)	-	0	0.77
*Staphylococcus*	42 (56.75%)	-	-	11 (26.19%)	0.4	0.0001
*Streptococcus*	3 (60.00%)	-	2 (66.66%)	-	0.2	0.04
*Streptococcus*	3 (60.00%)	-	-	1 (33.33%)	0.2	0.01
*Neisseria méningitidis*	1 (100%)	-	-	1 (100%)	0.1	0.12
**2020**						
*Enterobacteriaceae*	60 (86.95%)	23 (38.33%)	-	-	0.7	0.0001
*Staphylococcus*	22 (45.83%)	-	3 (13.63%)	-	−0.2	0.85
*Staphylococcus*	22 (45.83%)	-	-	5 (22.72%)	0.8	0.0001
*Streptococcus*	2 (60.00%)	-	1 (50.00%)	-	0.2	0.13
*Streptococcus*	2 (60.00%)	-	-	1 (50.00%)	0.1	0.58
**2021**						
*Enterobacteriaceae*	61 (64.21%)	31 (50.81%)	-	-	0.5	0.0001
*Staphylococcus*	23 (48.93%)	-	-	12 (52.17%)	0.7	0.0001
*Streptococcus*	2 (50.00%)	-	1 (50.00%)	-	0.2	0.16
*Streptococcus*	2 (50.00%)	-	-	1 (50.00%)	0.4	0.0005

(-): non available.

## Data Availability

All data are available upon request.

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
