# Peer review of "Prevalence and Antibiotic Resistance Pattern of Streptococcus, Staphylococcus, Neisseria meningitidis and Enterobacteriaceae in Two Reference Hospitals of Yaoundé: An Overview before and during COVID-19 Pandemic Era"

_antibiotics, 2023, doi:10.3390/antibiotics12050929_

Round 1

Reviewer 1 Report

The topics is extremely actual. 

I recommend a few modification.

Double check the English.

Please describe the abbreviations when you use them for the first time

The  first part of abstract need to be rewrite!

The title and the article should be reformat based on new classification ESBL-producing Enterobacterales | HAI | CDC

I consider that article should be restructure, the number of samples is limited 426, comparing with my hospital we process between 500 - 700 samples per day! expand the time and number of sample analized!

The conclusion need to clear and specific. My recommendation is to focus on short conclusion which can be usefull for clinicians.

Please recheck the References order.

Author Response

The topics is extremely actual. I recommend a few modification.

Answer: Thanks dear reviewer

Double check the English.

Answer: Done dear reviewer. Thanks

Please describe the abbreviations when you use them for the first time

Answer: Ok dear reviewer. Thanks

The first part of abstract need to be rewrite!

Answer: Done dear reviewer. Thanks

The title and the article should be reformat based on new classification ESBL-producing Enterobacterales | HAI | CDC

Answer: Thanks dear reviewer.

I consider that article should be restructure, the number of samples is limited 426, comparing with my hospital we process between 500 - 700 samples per day! expand the time and number of sample analized!

Answer: Dear reviewer. Thanks for your remark. This point is actually context-dependant. In fact, in our context people generally come to hospital when they haven’t found better solution with traditional medicine, self-medications and/or herbal remedies. And this phenomenon was compound during the COVID-19 pandemic as people were afraid to getting infected in Hospitals.

The conclusion need to clear and specific. My recommendation is to focus on short conclusion which can be usefull for clinicians.

Answer: Many thanks dear reviewer.

Please recheck the References order.

Answer: Thanks dear reviewer

Reviewer 2 Report

The study is interesting and relevant to the field. The authors showed the prevalence and antibiotic resistance pattern of bacteria isolated in two referral health facilities in Yaounde before and during the COVID-19 pandemic era. However, some key controls are missing and some revisions are needed in the text.

 1-      Figure 1. Authors must correct the writing of the genus and species in italics and only the genus with the first letter capitalized.

 2-      Line 324-327 ... “but greater resistance burden with the lowest bacteria amount and peak of bacteria resistance registered in 2020, the year of COVID-19”... The results in the text contrast with this statement.  In Figure 3, data showed a reduction in the resistance rate for the Staphylococcus genus of ~10% between 2019 and 2020.

 3-      Table II. Change "Global resistant rate" to Resistance rate...

 4-      Explain in the legend of Tables III and V the meaning of the symbol (-). I was in doubt if the meaning was unrealized, zero, susceptibility...

 5-      Azithromycin is a synthetic derivative of erythromycin (macrolides), whose mechanism of action is to inhibit protein synthesis by acting on the 50S subunit. Authors should explain the observed difference between the same bacterial genera for the two macrolides, as shown in table V.

 6-      Lines 272 and 273 repeat the data collection location.

 7-      Lines 279-280. The authors described that the results found are in accordance with the previous publication by Chih-Cheng et al (2021), who verified a decrease in Escherichia coli strains. However, Figure 1 shows an increase in E. coli isolates in 2020 compared to 2019.

 8-      Line 331- correct "Of" to “of” and correctly cite the Ruwandi et al (2022) reference.

Author Response

Comments and Suggestions for Authors

The study is interesting and relevant to the field. The authors showed the prevalence and antibiotic resistance pattern of bacteria isolated in two referral health facilities in Yaounde before and during the COVID-19 pandemic era. However, some key controls are missing and some revisions are needed in the text.

Answer: Many thanks dear reviewer

  • Figure 1. Authors must correct the writing of the genus and species in italics and only the genus with the first letter capitalized.

Answer: Many thanks dear reviewer

  • Line 324-327 ... “but greater resistance burden with the lowest bacteria amount and peak of bacteria resistance registered in 2020, the year of COVID-19”... The results in the text contrast with this statement.  In Figure 3, data showed a reduction in the resistance rate for the Staphylococcusgenus of ~10% between 2019 and 2020.

Answer: Thanks dear reviewer, we revised it

  • Table II. Change "Global resistant rate" to Resistance rate...

Answer: Thanks dear reviewer, we changed it

  • Explain in the legend of Tables III and V the meaning of the symbol (-). I was in doubt if the meaning was unrealized, zero, susceptibility...

Answer: Done dear reviewer, Thanks. In fact, the symbol (-) means “reference” in table II and “non available” in table V.

  • Azithromycin is a synthetic derivative of erythromycin (macrolides), whose mechanism of action is to inhibit protein synthesis by acting on the 50S subunit. Authors should explain the observed difference between the same bacterial genera for the two macrolides, as shown in table V.

Answer: Done dear reviewer. Indeed, antimicrobial resistance is described to be multifactorial. We strongly believe that the over and misuse of Azithromycin nowadays as compared to erythromycin in our context (even far before the COVID-19 onset) might be some factors.

  • Lines 272 and 273 repeat the data collection location.

Answer: Thanks again dear reviewer, we corrected it.

  • Lines 279-280. The authors described that the results found are in accordance with the previous publication by Chih-Cheng et al (2021), who verified a decrease in Escherichia coli However, Figure 1 shows an increase in E. coliisolates in 2020 compared to 2019.

Answer: Thanks again dear reviewer. We revised it

  • Line 331- correct "Of" to “of” and correctly cite the Ruwandi et al (2022) reference.

Answer: Done dear reviewer. Thanks

Reviewer 3 Report

Djuikoue et al, performed a study to determine the prevalence and antibiotic resistance pattern of bacteria isolated in two referral health facilities in Yaounde before and during the COVID-19 pandemic era. The aim of this study was to determine the prevalence and antibiotic resistance pattern of bacteria isolated in two referral health facilities in Yaounde before and during the COVID-19 pandemic era.

It is a well-known phenomenon that the frequent use of antibiotics increases MDR.

No new insight in this manuscript that sounds like adding value to existing literature.

There are fundamental errors in reporting and analyzing results. The methods lack clarity, Bacterial identification, and Antibiogram testing, which standards either CLSI or EUCAST were followed to determine the level of resistance. There is a clear scientific misunderstanding. This paper lacks scientific vigor.

The manuscript is poorly written with materials/methods and results presented in an ambiguous manner, and massive revision is needed to publish this work in a Journal of high impact such as Biomolecules MDPI.

The following are the two main reasons why this paper should not be accepted for publication in its current form.

1.       The procedures and/or analysis of the data is seen to be defective. The results are conflicting.

2.       The conclusions cannot be justified based on the data presented.

Below are suggested points that will be helpful to improve the quality of the current manuscript.

Major Revisions:

·         Rewrite the whole paper including the scientific methods used in this study in detail and analyze the result carefully and scientifically.

·         Use the correct terminology like AMR and MDR

Minor Revisions:

·         Line39 : bacteria Stains should read bacteria Strains

·         Line50Monitor should read as monitored

·         Line 79: Multidrug resistant (BMR) bacteria should read as (MDR) here and throughout, BMR never heard of , It should be AMR, MDR most commonly used

·         Line 82 enterobacteriaceae should be Enterobacteriaceae

·         Line 87: methicylline should read as Methicillin

·         Line 98: community infectiology, should read as community infections

·         Line 99: nosocomial infectiology, same as above

·         Line 101: Bacteria are talked to be multi-resistant to antibiotics (BMR), edit this please like considered to be MDR

·         Line 133: 120in 2020 need a space

·         Line 146: Table 1 (N= should be small letter “n” also the ] used is reverse orientation

·         Line 257: R and P-value are written incorrectly

Author Response

Comments and Suggestions for Authors

Djuikoue et al, performed a study to determine the prevalence and antibiotic resistance pattern of bacteria isolated in two referral health facilities in Yaounde before and during the COVID-19 pandemic era. The aim of this study was to determine the prevalence and antibiotic resistance pattern of bacteria isolated in two referral health facilities in Yaounde before and during the COVID-19 pandemic era.

It is a well-known phenomenon that the frequent use of antibiotics increases MDR. No new insight in this manuscript that sounds like adding value to existing literature.

Answer: Thanks very much dear reviewer for your remark. As you know COVID-19 onset has increased the rate of over and misuse of potent antibiotics in our context in low and middle income countries and we found interesting to present the state of the situation concerning the resistance (AMR) of common bacteria genera to some selected potent antibiotics before and during the pandemic Era. We strongly believe that such a report remains crucial for rapid preparedness and response to AMR for the future as we are probably running out of the COVID-19 pandemic era already.

There are fundamental errors in reporting and analyzing results. The methods lack clarity, Bacterial identification, and Antibiogram testing, which standards either CLSI or EUCAST were followed to determine the level of resistance. There is a clear scientific misunderstanding. This paper lacks scientific vigor.

Answer: Thanks very much dear reviewer for those remarks. We took them into account in the corrected manuscript

The manuscript is poorly written with materials/methods and results presented in an ambiguous manner, and massive revision is needed to publish this work in a Journal of high impact such as Biomolecules MDPI.

The following are the two main reasons why this paper should not be accepted for publication in its current form.

  1. The procedures and/or analysis of the data is seen to be defective. The results are conflicting.

Answer: Thanks very much dear reviewer. We fully revised the data analysis

  1. The conclusions cannot be justified based on the data presented.

Answer: Thanks very much dear reviewer. We also reshaped it according to the results

Below are suggested points that will be helpful to improve the quality of the current manuscript.

Major Revisions:

  • Rewrite the whole paper including the scientific methods used in this study in detail and analyze the result carefully and scientifically.

Answer: Thanks very much dear reviewer. We did it accordingly

  • Use the correct terminology like AMR and MDR

Answer: Done dear reviewer. Thanks

Minor Revisions:

  • Line39 : bacteria Stains should read bacteria Strains

Answer: Thanks dear reviewer.

  • Line50Monitor should read as monitored

Answer: Thanks dear reviewer.

  • Line 79: Multidrug resistant (BMR) bacteria should read as (MDR) here and throughout, BMR never heard of , It should be AMR, MDR most commonly used

Answer: Thanks dear reviewer.

  • Line 82 enterobacteriaceae should be Enterobacteriaceae

Answer: Thanks dear reviewer

  • Line 87: methicylline should read as Methicillin

Answer: Thanks dear reviewer

  • Line 98: community infectiology, should read as community infections

Answer: Thanks dear reviewer

  • Line 99: nosocomial infectiology, same as above

Answer: Thanks dear reviewer

  • Line 101: Bacteria are talked to be multi-resistant to antibiotics (BMR), edit this please like considered to be MDR

Answer: Thanks dear reviewer

  • Line 133: 120in 2020 need a space

Answer: Thanks dear reviewer

  • Line 146: Table 1 (N= should be small letter “n” also the ] used is reverse orientation

Answer: Thanks dear reviewer

  • Line 257: R and P-value are written incorrectly

Answer: Thanks dear reviewer

Reviewer 4 Report

Dear authors,

Although this is a good paper, some modifications are required to improve the quality of this manuscript:

1. You must take into consideration an English editing of the manuscript. There are many mistakes, and many sentences are poorly written.

2. I don't think writing the name of the antibiotics in capitals is necessary.

3. Lines 79, 93, 101-> multidrug resistant needs to be abbreviated MDR. Also, you must be consistent; only the first time it appears in the text, you must write the full term.

3. Line 117 -> you must explain the terms DDD and DHD.

4. Lines 191, 192 -> please correct "bacterial resistance" and "lower rate of bacterial resistance."

5. Line 282 -> contrasting

6. Line 300 -> write with capitals Proteus mirabilis 

7. Lines 305-306 -> these lines need to be corrected.

8. You have some words written with capitals in the middle of the sentence. I don't think this is necessary, e.g., lines 305, 316, 331.

9. Line 333 -> write with capital the author's name.

10. Line 346, 359, 361 - > modify in italics the name of the bacterium

11. Line 353 -> you must use the singular form, "bacterium."

12. Table 1. Please verify all the typos.

13. Figure 1 -> I don't think it is necessary to write the full names of the bacteria in the capitals.

14. All the manuscript requires extensive editing regarding typos and English; there are many writing errors.

15. Line 303 -> In the text, references numbers should be placed in square brackets [ ] and placed before the punctuation; for example, [1], [1-3], or [1,3]. Please verify the manuscript.

16. Line 365 -> Please correct this sentence.

17. The conclusions are very scarce and already well-known, and there are no insights into potential solutions.

18. The reference list does not respect the requirements. Please verify.

Author Response

Dear authors,

Although this is a good paper, some modifications are required to improve the quality of this manuscript:

Answer: Many Thanks dear reviewer

  1. You must take into consideration an English editing of the manuscript. There are many mistakes, and many sentences are poorly written.

Answer: Done dear reviewer. Thanks

  1. I don't think writing the name of the antibiotics in capitals is necessary.

Answer: Thanks dear reviewer. We have corrected

  1. Lines 79, 93, 101-> multidrug resistant needs to be abbreviated MDR. Also, you must be consistent; only the first time it appears in the text, you must write the full term.

Answer: Many Thanks dear reviewer

  1. Line 117 -> you must explain the terms DDD and DHD.

Answer: Done dear reviewer. Thanks

  1. Lines 191, 192 -> please correct "bacterial resistance" and "lower rate of bacterial resistance."

Answer: Done dear reviewer. Thanks

  1. Line 282 -> contrasting

Answer: Done dear reviewer. Thanks very much

  1. Line 300 -> write with capitals Proteus mirabilis 

Answer: Done dear reviewer. Thanks very much

  1. Lines 305-306 -> these lines need to be corrected.

Answer: Done dear reviewer. Thanks

  1. You have some words written with capitals in the middle of the sentence. I don't think this is necessary, e.g., lines 305, 316, 331.

Answer: corrected dear reviewer. Thanks

  1. Line 333 -> write with capital the author's name.

Answer: corrected dear reviewer. Thanks

  1. Line 346, 359, 361 - > modify in italics the name of the bacterium

Answer: corrected dear reviewer. Thanks

  1. Line 353 -> you must use the singular form, "bacterium."

Answer: corrected dear reviewer. Thanks very much

  1. Table 1. Please verify all the typos.

Answer: done dear reviewer. Thanks

  1. Figure 1 -> I don't think it is necessary to write the full names of the bacteria in the capitals.

Answer: corrected dear reviewer. Thanks

  1. All the manuscript requires extensive editing regarding typos and English; there are many writing errors.

Answer: done dear reviewer. Thanks

  1. Line 303 -> In the text, references numbers should be placed in square brackets [ ] and placed before the punctuation; for example, [1], [1-3], or [1,3]. Please verify the manuscript.

Answer: done dear reviewer. Thanks

  1. Line 365 -> Please correct this sentence.

Answer: done dear reviewer. Thanks

  1. The conclusions are very scarce and already well-known, and there are no insights into potential solutions.

Answer: We revised it dear reviewer. Thanks

  1. The reference list does not respect the requirements. Please verify.

Answer: done dear reviewer. Thanks

Round 2

Reviewer 1 Report

Accept for publication!

Author Response

Many thanks dear Reviewer

Reviewer 3 Report

I am sorry but I am not still convinced the manuscript has the potential to be published in MDPI Journal.  It is not clear to me how the antibiogram assay was performed and which method was adopted. Just stating EUCAST recommendations were followed is not enough. No details were provided on how the susceptibility assay was performed and what method was used to ID the strains used in this study.

Author Response

I am sorry but I am not still convinced the manuscript has the potential to be published in MDPI Journal.  It is not clear to me how the antibiogram assay was performed and which method was adopted. Just stating EUCAST recommendations were followed is not enough. No details were provided on how the susceptibility assay was performed and what method was used to ID the strains used in this study.

Dear reviewer, many thanks once more to help taking this manuscript to a much better version. Taking into account your comment, we added a sub-section titled “Microbiological method” that describes how strains ID and antibiogram were retrospectively performed by the laboratory personnel of the bacteriology laboratory of the two selected reference hospitals in Cameroon.